# SeMv-3D: Towards Semantic and Mutil-view Consistency simultaneously for General Text-to-3D Generation with Triplane Priors

## Abstract

Recent advancements in generic 3D content generation from text prompts have been remarkable by fine-tuning text-to-image diffusion (T2I) models or employing these T2I models as priors to learn a general text-to-3D model. While fine-tuning-based methods ensure great alignment between text and generated views, i.e., **semantic consistency**, their ability to achieve multi-view consistency is hampered by the absence of 3D constraints, even in limited view. In contrast, prior-based methods focus on regressing 3D shapes with any view that maintains uniformity and coherence across views, i.e., **multi-view consistency**, but such approaches inevitably compromise visual-textual alignment, leading to a loss of semantic details in the generated objects. To achieve semantic and multi-view consistency simultaneously, we propose *SeMv-3D*, a novel framework for general text-to-3d generation. Specifically, we propose a Triplane Prior Learner (TPL) that learns triplane priors with 3D spatial features to maintain consistency among different views at the 3D level, e.g., geometry and texture. Moreover, we design a Semantic-aligned View Synthesizer (SVS) that preserves the alignment between 3D spatial features and textual semantics in latent space. In SVS, we devise a simple yet effective batch sampling and rendering strategy that can generate arbitrary views in a single feed-forward inference. Extensive experiments present our SeMv-3D's superiority over state-of-the-art performances with semantic and multi-view consistency in any view. Our code and more visual results are available at `https://anonymous.4open.science/r/SeMv-3D-6425`.

**Input:** *"Mario is wearing his signature red hat with a 'M' on it, blue overalls, white gloves, and brown shoes, with arms open."*

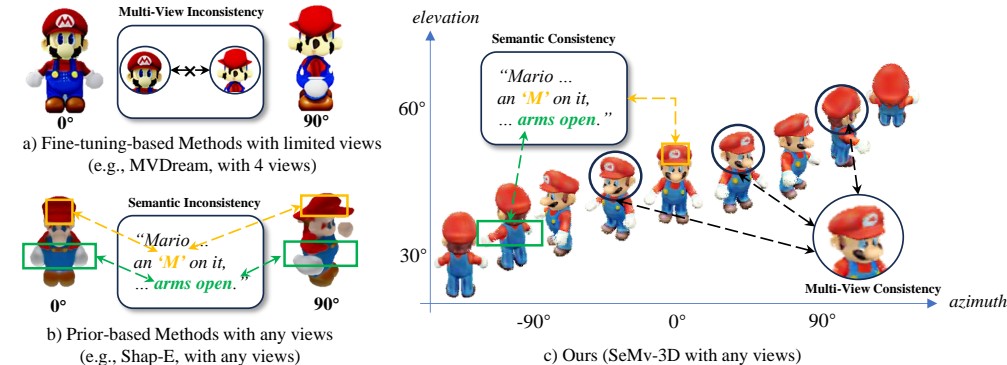

Figure 1: **Visual comparison with SOTA baselines and our SeMv-3D.** The two mainstream lines of general text-to-3d: a) Fine-tuning-based methods and b) Prior-based methods show two core challenges: multi-view inconsistency and semantic inconsistency, respectively. Our SeMv-3D c) can simultaneously maintain multi-view consistency and semantic consistency.

## 1 Introduction

Text-to-3D generation (T23D) aims to generate corresponding 3D content based on text prompts with a broad range of applications, including games, movies, virtual/augmented reality, and robotics.

The previous works mainly focus on a per-scene optimization problem (Poole et al., 2023; Lin et al., 2023; Wang et al., 2023; Chen et al., 2023), which yields fine texture and geometric details. However, these methods incur substantial time and computational overhead, as each object generated requires multiple optimizations to approximate the textual semantics. To overcome this issue, the general text-to-3D has been proposed by learning a generic model capable of synthesizing various objects in a feed-forward manner, which is a flexible and promising way. Without optimized refinement for pre-scene, general text-to-3D faces two core challenges: a) **Multi-view Consistency**, which maintains coherence across multiple 3D views, and b) **Semantic Consistency**, which requires semantic alignment of the generated 3D context with the text.

Benefiting from the great breakthroughs in the text-to-image diffusion (T2I) models, two research lines of rationalization have recently emerged in general text-to-3D, including fine-tuning T2I models and utilizing these models as priors to train 3D generation models. Specifically, fine-tuning-based methods seek to transfer the strong single-view generation capabilities of pretrained T2I models (e.g., great semantic alignment between text and vision) directly to generate multiple views with consistent relationships, such as MVDream (Shi et al., 2023) and DreamView (Yan et al., 2024). Yet these methods are inherently ambiguous without explicit 3D constraints, leading to notorious multi-view inconsistency (e.g., multi-face Janus problem, shown as Figure 1a) and limited-view. Conversely, prior-based methods primarily leverage the T2I models as semantic-visual initialization and subsequently train on large-scale 3D datasets. They solely focus on regressing the corresponding 3D shapes, which naturally ensures consistency across multiple views, such as Shap-E (Jun & Nichol, 2023) and VolumeDiffusion (Tang et al., 2023). However, it sacrifices portions of the well-learned semantic alignment information of the original T2I model, inevitably resulting in inconsistency between the generated visuals and their corresponding semantics, presented in Figure 1b. Thus, how to effectively and simultaneously achieve semantic and multi-view consistency remains to be explored for the general text-to-3D task.

Toward the above goal, we propose a novel framework, named *SeMv-3D*, which learns an efficient triplane prior to ensure uniformity across all views of an object and align its semantics with the text. Empirically, the triplane has been validated as an efficient and compact 3D representation for object modeling (Chan et al., 2022). Unlike existing methods that directly learn the entire triplane features, we emphasize spatial correspondence within the triplane to capture the underlying 3D details. Specifically, we propose a **T**riplane **P**rior **L**earner (**TPL**) that integrates 3D spatial features into a triplane prior. In practice, TPL first eliminates irrelevant backgrounds or components to preserve essential 3D information by our object retention module and then captures spatial correspondence within triplane space to enhance its visual coherence by triplane orthogonalization module, a new task-specific attention component. Moreover, we design **S**emantic-aligned **V**iew **S**ynthesizer (**SVS**) that deeply interacts between textual and visual features within triplane priors through a triplane latents transformation module, significantly improving semantic consistency. Additionally, in SVS, we incorporate a simple yet effective batch sampling and rendering strategy (by fitting multiple views at once), enabling the generation of any view in one single step. From Figure 1c, we can see that our method performs better in multi-view and semantic consistency than other compared methods.

To summarize, our main contributions are threefold:

1) We devise a *SeMv-3D*, a novel general text-to-3D framework, which simultaneously ensures semantic and multi-view consistency.

2) We propose a **TPL**, which learns a triplane prior to effectively capture consistent 3D features across generated views. Moreover, we devise a **SVS** that deeply explores the alignment between textual and 3D visual information, substantially improving semantic consistency.

3) Extensive experiments show the superiority of our *SeMv-3D* in both qualitative and quantitative terms of multi-view and semantic consistency. Besides, our method presents a new property, i.e., the generation of any view in one feed-forward inference.

## 2 RELATED WORKS

Text-to-3D (T23D) aims to synthesize 3D representations (3D voxels, point clouds, multi-view images, and meshes) from textual descriptions. Early works of T23D directly train generation models

on small-scale 3D datasets, which restricted the semantic diversity and geometry fidelity of the 3D outputs. With the emergence of pretrained Text-to-Image (T2I) diffusion models, recent works utilize semantic-visual prior knowledge of these T2I models for fine-grained and diverse 3D generation. Existing works can be grouped into two categories based on generalization ability: 1) Per-scene Text-to-3D and 2) General Text-to-3D.

**Per-scene Text-to-3D.** Per-scene Text-to-3D requires per-scene optimization when generating a new scene. The mainstream idea is using knowledge from pre-trained T2I models to guide the optimization of 3D representations. DreamFusion (Poole et al., 2023) employs a technique known as Score Distillation Sampling (SDS). This approach utilizes large-scale image diffusion models (Rombach et al., 2022; Saharia et al., 2022) to iteratively refine 3D models to match specific prompts or images. Similarly, ProlificDreamer (Wang et al., 2023) develops Variational Score Distillation (VSD), a structured variational framework that effectively reduces the over-saturation problems found in SDS while also increasing diversity. Further enhancements are offered by several studies (Qian et al., 2023; Qiu et al., 2023; Wang & Shi, 2023), which address the challenges of multiple faces by using diffusion models fine-tuningd on 3D data. The strategy of amortized score distillation is examined in other references (Lorraine et al., 2023; Qian et al., 2024). Numerous additional works (Chen et al., 2023; Lin et al., 2023; Tsalicoglou et al., 2023; Zhu & Zhuang, 2023) have substantially improved both the speed and quality of these approaches. Despite fine-grained texture details through optimization, these methods usually require a lengthy period, ranging from minutes to hours, to generate only a single object. Contrastly, our approach employs a feed-forward method that requires no per-scene optimization.

**General Text-to-3D.** Methods in General Text-to-3D achieve open-domain T23D without needing additional optimization for each new scene. These methods can be divided into two categories based on their implementation process: fine-tuning-based and prior-based approaches. Prior work SDFusion (Cheng et al., 2023) takes dense SDF grids as the 3D representation, which is computational cost and unable to render textures. Point-E (Nichol et al., 2022) and Shap-E (Jun & Nichol, 2023), trained on millions of 3D assets, generate point clouds and meshes respectively. 3DGen (Gupta et al., 2023) combines a triplane VAE for learning latent representations of textured meshes with a conditional diffusion model for generating the triplane features. VolumeDiffusion (Tang et al., 2023) trains an efficient volumetric encoder to produce training data for the diffusion model. With insufficient 3D data to learn, recent works tend to utilize 2D priors to help the training. Inspired by image-to-3D models (Liu et al., 2023b;a), image diffusion models are adopted for 3D generation. MVDream (Shi et al., 2023) and DreamView (Yan et al., 2024) attempt to jointly train the image generation model with high-quality normal images and limited multi-view object images to produce various object images. Recently, SPAD (Kant et al., 2024) builds upon MVDream to achieve arbitrary view generation. Despite these advancements, current methods still struggle to generate both semantic and multi-view consistent views. In contrast, our approach learns a complete 3D prior, enabling arbitrary view generation while maintaining consistent results across different views.

## 3 APPROACH

### 3.1 OVERVIEW

To simultaneously maintain semantic and multi-view consistency, we propose a novel framework called *SeMv-3D* for general Text-to-3D, which is illustrated in Figure 2. Our *SeMv-3D* generally consists of two core components: *Triplane Prior Learner* (TPL) and *Semantic-aligned View Synthesizer* (SVS). Given a textual description, TPL in Sec 3.2 first integrates the orthogonal correspondence in visual features to learn a consistent triplane prior. Based on the triplane prior in TPL, SVS in Sec 3.3 then transforms it into latent space while aligning it with semantics information and finally renders arbitrary views by incorporating a simple yet effective strategy in one single step.

### 3.2 TRIPLANE PRIOR LEARNER

3D representation constraints are crucial to ensure multi-view consistency. Especially, triplanes are acknowledged to be computationally efficient and effective 3D representations for characterizing 3D objects. However, directly regressing to triplane features like previous works will neglect detailed visual correspondence among views. Consequently, to achieve both efficient 3D representation con-

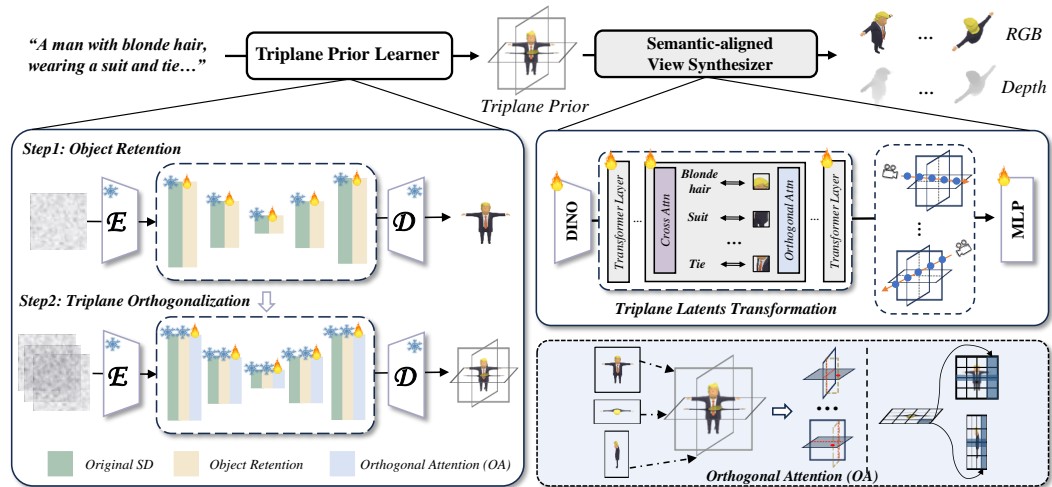

Figure 2: **The overall framework of SeMv-3D.** SeMv-3D consists of two components: 1) Triplane Prior Learner (**TPL**) that learns a triplane prior to capture consistent 3D visual details and 2) Semantic-aligned View Synthesizer (**SVS**) that enhances the alignment between the semantic with 3D content and enables single-step generation of arbitrary views. Here, Orthogonal Attention (**OA**) focuses on the orthogonal correspondences within the triplane, maintaining triplane consistency and extracting fine-grained features.

straints and fine-grained multi-view consistency, we propose Triplane Prior Learner (TPL) as shown in the left part of Figure 2, which models detailed spatial correspondence in objects into a triplane prior. More illustration information is provided in Appendix. A.3.

Specifically, the TPL takes textual descriptions $T$ as inputs and outputs a triplane prior $P$, which can be formalized as $P_{tri} = TPL(T)$. The mapping of $TPL(\cdot)$ is built upon a powerful pretrained T2I model SD2.1 (Rombach et al., 2022) for utilizing 2D priors. To preserve original T2I knowledge, we freeze the T2I model and add new learnable parameters for training our TPL. The training process is disentangled into two subsequent steps: Object Retention in 3.2.1 and Triplane Orthogonalization in 3.2.2.

### 3.2.1 OBJECT RETENTION

Current pretrained T2I models are able to produce images of high quality and great details. However, we only focus on the main object and need no other stuff like background. In the context of such diverse generative capabilities, directly fine-tuning would be severely impacted by irrelevant information, making it difficult to learn triplane effectively. Therefore, to retain the main object of interest while removing unnecessary elements, we introduced an Object Retention (OR).

Specifically, we add the additional parameters $\theta_{OR}$ and train the newly added parameters on a text-object dataset with the object images' background removed. In practice, one residual block and one attention block are plugged into each level of the UNet network before upscale and downscale, while all other pre-trained layers are frozen during training. The learning objective function can be described as follows:

$$\mathcal{L}_{\text{OR}} = \mathbb{E}_{t\sim[1,T],\mathbf{x}_0,\boldsymbol{\epsilon}_t}\left[\|\boldsymbol{\epsilon}_t - \boldsymbol{\epsilon}_{\theta_{\text{OR}}}(\sqrt{\bar{\alpha}_t}\mathbf{x}_0^i + \sqrt{1-\bar{\alpha}_t}\boldsymbol{\epsilon}_t, t, c)\|^2\right], \tag{1}$$

where $\epsilon_t$ is the added noise for diffusion process for the timestep $t$ on the condition text prompt $c$, $\bar{\alpha}_t$ is the pre-defined hyper-parameters for the sampling scheduler, and $\mathbf{x}_0^i$ is a clean object image sampling from random viewpoints.

### 3.2.2 TRIPLANE ORTHOGONALIZATION

After the Object Retention training, our model retains the strong capability to generate only the primary object. Next, to learn spatial orthogonal relationships within triplane priors, we introduce the Triplane Orthogonalization (TO) module. Similarly, we increase the learning parameters $\theta_{TO}$

and train on a dataset where the front, top, and side views—completely orthogonal perspectives—are selected as the ground truth for the triplane.

In practice, we append a TO module subsequent to each OR module. During the triplane learning, we freeze all other components and only optimize added TO modules with the triplane supervision, whose objective function can be expressed as follows:

$$\mathcal{L}_{\text{TO}} = \mathbb{E}_{t\sim[1,T],\mathbf{x}_0,\boldsymbol{\epsilon}_t} \sum_{i\in\{xy,xz,yz\}} \left[ \|\boldsymbol{\epsilon}_t^i - \boldsymbol{\epsilon}_{\theta_{\text{TO}}}(\sqrt{\bar{\alpha}_t}\mathbf{x}_0^i + \sqrt{1-\bar{\alpha}_t}\boldsymbol{\epsilon}_t^i, t, c)\|^2 \right]. \quad (2)$$

However, directly training $\theta_{TO}$ to regress the triplane prior (front, top, and side images) leads to a degradation of the spatial correspondence between different views. To address this issue, existing works (Shi et al., 2023; Blattmann et al., 2023) introduce temporal attention, which establishes a relationship among pixels in different views, to learn the correspondence of multi-views. Nevertheless, temporal attention is not effective in handling our sparse triplanes with significant pixel deviation between neighboring views. Under such large shifts, temporal attention can only grasp a rough triplane relationship and fails to capture the spatial correspondence and consistency within each plane (demonstrated in Fig.4).

To this end, we propose our **orthogonal attention** (OA), which focuses on the orthogonal spatial relationship between triplanes and correlates the orthogonality to ensure consistency, **as shown in Fig. 2**. For example, given a pixel $(a, b, -)$ in the latent $xy$-plane which needs to focus on pixels in the other two orthogonal planes, it should intersect all pixels with the same x-axis coordinate $(a)$ in the $xz$-plane and all pixels with the same y-axis coordinate $(b)$ in the $yz$-plane, and pixels on the cross line between the corresponding planes. The orthogonal attention can be expressed as follows:

$$\begin{aligned}
\text{OA}(\mathbf{P}_{xy}|\mathbf{P}_{xz}, \mathbf{P}_{yz}) &= \text{OA}_x(\mathbf{P}_{xy}, \mathbf{P}_{xz}) + \text{OA}_y(\mathbf{P}_{xy}, \mathbf{P}_{yz}), \\
\text{OA}(\mathbf{P}_{xz}|\mathbf{P}_{xy}, \mathbf{P}_{yz}) &= \text{OA}_x(\mathbf{P}_{xz}, \mathbf{P}_{xy}) + \text{OA}_z(\mathbf{P}_{xz}, \mathbf{P}_{yz}), \\
\text{OA}(\mathbf{P}_{yz}|\mathbf{P}_{xz}, \mathbf{P}_{xy}) &= \text{OA}_z(\mathbf{P}_{yz}, \mathbf{P}_{xz}) + \text{OA}_y(\mathbf{P}_{yz}, \mathbf{P}_{xy}),
\end{aligned} \quad (3)$$

and

$$\text{OA}_i(\mathbf{P}_1, \mathbf{P}_2) = \prod_{M\in\mathbf{P}_1} \text{softmax}\left(\frac{W_Q(M)W_K(N)^T}{\sqrt{d_{W_K(N)}}}\right) W_V(N), \text{ s.t.i } i\in\{x,y,z\}, \quad (4)$$

where

$$M = \{K|K\in\mathbf{P}_1\},\ N =\{K|K\in\mathbf{P}_2\ \&(Coord_i(M) = Coord_i(K)\mid K\in(\mathbf{P}_1\cap\mathbf{P}_2))\}, \quad (5)$$

$P_i$ represents the i-th plane in triplane, $W_Q$, $W_K$, and $W_V$ refer to query, key, and value mapping functions, and $Coord_i(\cdot)$ indicates the i-axis coordinate.

### 3.3 SEMANTIC-ALIGNED VIEW SYNTHESIZER

Given the learned consistent triplane prior through our TPL, we aim to utilize it to synthesize multi-views. While current prior-based methods suffer from the sacrifices of well-learned textual-visual alignment in regressing 3D. To this end, we introduce a Semantic-aligned View Synthesizer (SVS) composed of a Triplane Latents Transformation module, in Sec 3.3.1, aiming to facilitate the deep interaction between textual and visual features to improve semantic consistency. While existing methods can only generate limited views or multi-view with multiple inference steps, we adopt a simple yet effective training strategy to generate arbitrary views in a single step, illustrated in Sec 3.3.2. More illustration information is presented in Appendix. A.3.

#### 3.3.1 TRIPLANE LATENTS TRANSFORMATION

The Triplane Latents Transformation (TLT) module plays a crucial role in SVS, learning the actual implicit triplane representation and further aligning semantics with orthogonalized 3D features. Unlike prior-based methods, which do not incorporate semantic alignment during the formation of implicit fields, our approach introduces semantic alignment during the construction of the triplane implicit field. Given the spatial orthogonality of the triplane, we do not simply incorporate text embeddings but instead align semantic features with the orthogonalized 3D triplane features. This

approach enables precise semantic matching across different 3D visual feature regions. To raise an example, *"blonde hair"* features could align with their visual features within orthogonalized triplanes.

In practice, we first extract the visual features from triplane prior $P$ via $DINO(\cdot)$ (Caron et al., 2021), denoted as $Token_{tri} = DINO(P)$. These features are then enriched with semantics $T$ through CA, represented as $CA(Token_{tri}, T)$. Through OA, we enable spatial orthogonal interactions of these semantically rich features, $OA(CA(Token_{tri}, T))$, thereby establishing finer-grained associations between 3D visual feature regions and semantic representations. During training, we transfer the processed features to the radiance field by $Transformer(\cdot)$, obtaining triplane latents $f_{Tri}$ that can be easily understood by the synthesizer and contain ample semantics and 3D information:

$$f_{Tri} = Transformer(OA(CA(Token_{tri}, T))) \tag{6}$$

### 3.3.2 Batch Sampling & Rendering

The batch sampling and rendering strategy is simple yet effective, designed to enable the generation of any views in one single feed-forward step. Following the (Chan et al., 2022; Mildenhall et al., 2020), we employ the triplane latents $f_{Tri}$ as implicit fields for ray sampling and rendering. In ray sampling, given a batch of camera positions $\mathbf{o}$, for a ray path $\mathbf{r}(t) = \mathbf{o}_i + t\mathbf{d}$ in the direction $\mathbf{d}$ that forms a pixel, we now will form a batch of pixels from different views. Then for each ray, we sample several points on it, where the sampling range is restricted by a near bound $t_n$ and a far bound $t_f$. Next, we calculate the three projected points on the triplane and concatenate their features to represent each sampled point with feature $\mathbf{f}(\mathbf{r}(t))$. Typically, for those projected points without integer coordinates, we interpolate the features from the four nearest pixels to obtain their representations. Finally, we accumulate all these sampled points to calculate the rendered pixels in a batch.

Specifically, we learn two MLP functions (i.e., $S$ and $C$) to predict the density $\sigma$ and color $\mathbf{c}$ of each point, as follows:

$$\begin{aligned} \sigma(\mathbf{r}(t)) &= S(\mathbf{r}(t), \mathbf{f}(\mathbf{r}(t))), \\ \mathbf{c}(\mathbf{r}(t)) &= C(\mathbf{r}(t), \mathbf{f}(\mathbf{r}(t))), \end{aligned} \tag{7}$$

Then, we calculate the pixel information accumulating all samples points as follows:

$$\mathbf{Pix}_{rgb} = \int_{t_n}^{t_f} T(t) \cdot \sigma(\mathbf{r}(t)) \cdot \mathbf{c}(\mathbf{r}(t)) \, dt, \tag{8}$$

where

$$T(t) = exp(-\int_{t_n}^{t} \sigma(\mathbf{r}(s)) \, ds). \tag{9}$$

Typically, RGB pixels can be totally discretely rendered for optimization since they are independent. In our experiments, a batch rendering strategy is employed to generate multiple views in a single step. With all pixel colors $\mathbf{Pix}_{rgb}$ in a batch calculated, we can obtain the batch images $\mathbf{I}$. Similarly, we can also obtain the corresponding masks $\mathbf{M}$ and depths $\mathbf{D}$. The object function can be expressed as follows:

$$\mathcal{L}_{Render} = \sum_{i=1}^{N}(\|\mathbf{I}^i - \mathbf{I}_{GT}^i\|_2 + \lambda_M\|\mathbf{M}^i - \mathbf{M}_{GT}^i\|_2 + \tag{10}$$

$$\lambda_D\|\mathbf{D}^i - \mathbf{D}_{GT}^i\|_2 + \lambda_{lpips}(\mathcal{L}_{lpips}(\mathbf{I}^i, \mathbf{I}_{GT}^i)),$$

where $N$ indicates the view number used for training, and $\mathbf{I}_{GT}$, $\mathbf{M}_{GT}$ and $\mathbf{D}_{GT}$ refer respectively to the ground truth in pixel, mask and depth. $\mathcal{L}_{lpips}$ (Zhang et al., 2018) is the perceptual loss for better optimization. We set $\lambda_M = 0.5$, $\lambda_D = 1$, $\lambda_{lpips} = 2$ to balance each item.

## 4 Experiments

In this section, we conduct comprehensive experiments to evaluate our general text-to-3D framework, SeMv-3D, and provide comparative results against various baseline models. We first present qualitative comparisons with fine-tuning-based and prior-based methods in Sec. 4.2. Then We showcase quantitative comparisons based on objective metrics in Sec. 4.3 and subjective assessments from

a user study in Sec. 4.4. Finally, we carry out ablation studies to further demonstrate the efficiency of our framework design in Sec. 4.5. More visualizations and detailed analysis are provided in the Appendix A.2.

## 4.1 EXPERIMENT SETUP

**Evaluation Metrics.** Following previous work (Hong et al., 2024; Shi et al., 2023), we conduct a comprehensive evaluation incorporating both objective and subjective assessments. More details are provided in Appendix A.4.2. For *objective evaluation*, we select three commonly used evaluation metrics, including 1) **Clip Score** (Zhengwentai, 2023): which measure the consistency of the semantic alignment between the input text and generated object; 2) **Aesthetic Score**[1]: which represents the aesthetic performance of the generated object; and 3) **Views / One-step**: which indicates the upper limit on the number of views that the model can generate in one feed-forward step. For *subjective evaluation*, we conduct a user study in which users evaluate the results from three perspectives - 1) **Users Prefer**: similar to Aesthetics Scores, which indicates the user's liking for the generated views; 2) **Semantic Consistency**: which measures how well the generated objects match the text like Clip Scores; and 3) **Multi-view Consistency**: which assess the consistency of objects between each view.

**Baselines.** To showcase the outstanding performance of our SeMv-3D in both semantic and multi-view consistency, we also compare it with many state-of-the-art methods, which can be categorized into two types: **1) Fine-tuning-based methods** - MVDream (Shi et al., 2023) and DreamView (Yan et al., 2024) that generate high-quality but limited multi-views while SPAD (Kant et al., 2024) can generate any multi-view but with low consistency. **2) Prior-based methods** - (i) Point-E (Nichol et al., 2022) that employs DALLE (Ramesh et al., 2021) as priors and converts it into vivid point clouds. (ii) Shap-E (Jun & Nichol, 2023) that generates higher quality mesh representations based on the Point-E. (iii) VolumeDiffusion (Tang et al., 2023) that designs a volumetric encoder to produce various volumes. (iv) 3DTopia (Hong et al., 2024) that learns triplane features for further optimization. Particularly, we compare with these methods in a general text-to-3d setting, i.e., using inference only without any additional optimization or refinement to ensure fairness. Our proposed method belongs to the second category.

## 4.2 QUALITATIVE COMPARSION

**Comparison with Fine-tuning-based Methods.** Figure 3a shows the visualized comparison between our method and fine-tuning-based methods. From the figure, we observe that compared to fine-tuning-based methods without keeping multi-view consistency, our approach displays the strong capabilities of multi-view consistency and semantic consistency. Specifically, for some symmetrical objects, such as *"Mug"* and *"Car"*, fine-tuning-based methods can maintain the consistency of the main components, while for some localized areas, it cannot maintain the consistency, such as the handle and the color, as shown in Figure 3a (i) and (ii), respectively. Moreover, for texture-rich objects like the *"Cassette Player"*, MVdream and DreamView also lose complex textures across different views while SPAD shows nearly different object across views, illustrated in Figure 3a (iii). In contrast, our approach is unchanged between views in both overall and local details through the constraints of 3D triplane, maintaining good consistency. These results clearly prove the superiority of our SeMv-3D.

**Comparison with Prior-based Methods.** Figure 3b showcases qualitative comparison with state-of-the-art prior-based methods. In this experiment, we pick 6 views (at 60° intervals of azimuth angle) under their respective default settings (e.g., different elevations) with the optimal performance to ensure fairness. From the figure, we can see that, as previously stated, the prior-based methods are constrained in terms of semantic consistency. For example in Figure 3b (i), in terms of detail information, such as attributes (e.g., single, high top), the compared methods struggle to generate accurate semantics. Furthermore, for the total information, only Point-E enables to produce the *"ear cups"* and these methods can not accurately generate the *"headband"*, depicted in Figure 3b (ii). Conversely, our approach performs well to align the generated objects with the textual semantic, i.e., semantic consistency. Besides, our approach achieves higher fidelity, such as texture and geometry. These results further emphasize the effectiveness of our approach.

---

[1]https://github.com/grexzen/SD-Chad

(a) Comparison with Fine-tuning-based Methods

(b) Comparison with Prior-based Methods

Figure 3: **Performance comparison of Text-to-3D generation between baselines and our method (SeMv-3D) in qualitative aspect. a)** indicates our method achieves better multi-view consistency and comparable quality than the Fine-tuning-based Methods while **b)** shows our method maintains better semantic alignment with any-view than Prior-based Methods. More results are presented in the Appendix A.2.

### 4.3 QUANTITATIVE COMPARISON

The left of Table 1 lists the quantitative comparisons between baselines and our method SeMv-3D. Note that the clip score and aesthetic score only evaluate the front view generated by feeding the same 25 prompts into each method. From the table, we can find that: (i) Our method achieves an outstanding second-place Clip Score, 30.26, surpassing all similar prior-based methods and outperforming fine-tuning-based approaches such as MVDream and SPAD. This demonstrates that our approach achieves semantic consistency and generation quality comparable to the state-of-the-art, highlighting its strong competitiveness. (ii) Although our method does not achieve the highest aesthetic score, it still attains the best performance among prior-based methods and surpasses the latest fine-tuning-based approach, SPAD. This strongly demonstrates the exceptional effectiveness of our method. (iii) Compared with the existing baselines, our SeMv-3D can obtain arbitrary views of objects at once by our proposed batch sampling&rendering. In particular, the leading counterparts, MVDream and DreamView, generate only 4 views, which are far fewer than what our model can produce. This result highlights the powerful generative capability of our method.

Table 1: **Performance comparison with the state-of-the-art methods in the quantitative (left) and user study (right) aspects.**

| Methods | Quantitative Comparison | | | User Study | | |
|---|---|---|---|---|---|---|
| | Clip Scores | Aesthetic Scores | Views/ One-Step | Users Prefer | Semantic Consistency | Multi-view Consistency |
| MVDream | 30.09 | **4.8392** | 4 | 23.0% | 17.4% | 12.2% |
| DreamView | **31.57** | 4.73 | 4 | 19.0% | 19.2% | 10.2% |
| SPAD | 29.42 | 4.34 | **Any** | 13.2% | 15.8% | 9.3% |
| Point-E | 23.43 | 3.8603 | 1 | 1.0% | 1.6% | 2.0% |
| Shap-E | 28.90 | 4.3756 | 1 | 11.5% | 11.1% | 24.4% |
| VolumeDiffusion | 23.51 | 4.2969 | 1 | 0.3% | 0.3% | 1.2% |
| 3DTopia | 25.87 | 3.6202 | 1 | 1.6% | 6.3% | 4.1% |
| SeMv-3D (ours) | 30.26 | 4.4302 | **Any** | **29.6%** | **28.5%** | **36.6%** |

## 4.4 USER STUDY

To further validate the quality of our method, we conduct a user study on all methods. More details are provided in Appendix A.4.2. As illustrated in the right of Table 1, on average, 29.6% users prefer our model over others, meaning that our model is preferred over the best model of all baselines in most cases. Moreover, our model also achieves the best scores in terms of semantic consistency, and view consistency, reaching 28.5% and 36.6% of user preference. The above results further highlight the benefits of our approach to achieve semantic and multi-view consistency simultaneously.

## 4.5 ABLATION STUDY

In this section, we conduct comprehensive ablation studies to validate the effectiveness of each component in SeMv-3D, including Triplane Prior Learning (TPL) and Semantic-aligned View Synthesizer (SVS). More results are presented in Appendix A.2.3.

**Ablation Study of TPL.** Figure 4 evaluates the effectiveness of each component in TPL by taking successively our proposed Object Retention (OR) modules, Triplane Orthogonalization (TO) modules, and Orthogonalization Attention (OA) into the base model. Here, we select SD2.1 as our base model. Overall, all the proposed components contribute significantly to the total quality of generation. Specifically, the base model first performs the worst with accompanied by much irrelevant information. By integrating the OR into the baseline, extraneous backgrounds can be removed while retaining the subject well. It reveals the importance of OR, which avoids the influence of irrelevant information on the quality of generation. Then, the TO with adopting temporal attention is added to the above model, which aims to learn the orthogonal spatial relationships of the triplane. Unfortunately, temporal attention can only capture the general spatial correspondences, failing to preserve and align the finer details of the object itself. Finally, fusing the three modules into the base model strictly ensures the correct spatial relationships between the generated triplanes and effectively learns high-fidelity visual information with consistent multi-view alignment. The results indicate OA is more effective in grasping spatial correspondences than temporal attention.

**Ablation Study of SVS.** Figure 5a investigates the efficacy of SVS, which has core components, including Orthogonalization Attention (OA) and Cross-Attention (CA). Firstly, we find that removing OA drastically decreases quality in geometry and texture (e.g., decline of red cubes and unshaped pillows), which indicates that OA of SVS plays an important role in grasping high-precision detailed visual features from triplane prior. Then, both CA and OA are deleted, and in the absence of semantic guidance, the features extracted by the model from the triplane are somewhat biased, and useless features may be extracted to generate 3D. For example, the large out-of-shape artifacts at the end/head of the bed, suggesting that CA indeed serves as a semantic guidance. Finally, the final SVS can reconstruct multi-view outputs with realistic geometric details and consistent alignment across different views, demonstrating its efficacy.

**Generalization of SeMv-3D.** To further explore the generalization of our method, we conduct an experiment by reconstructing text input to generate different 3D content via SVS while maintaining the same triplane prior from TPL. The text is reconstructed in terms of local details, including

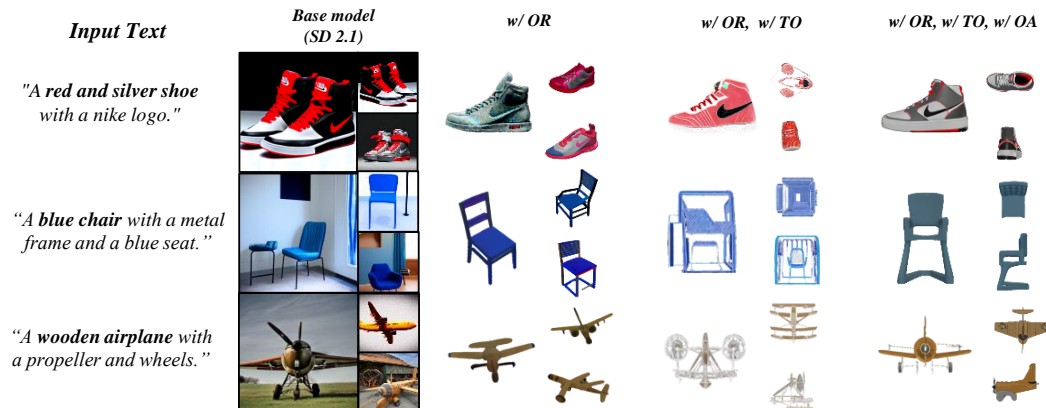

Figure 4: **Ablation study of the proposed modules in Triplane Prior Learning,** including 1) Object Retention (OR) that preserves essential 3D objects without backgrounds, Triplane Orthogonalization (TO) that tends to learn the orthogonal triplane relationships, and Orthogonalization Attention (OA) that maintains consistent and great 3D details in geometry and texture.

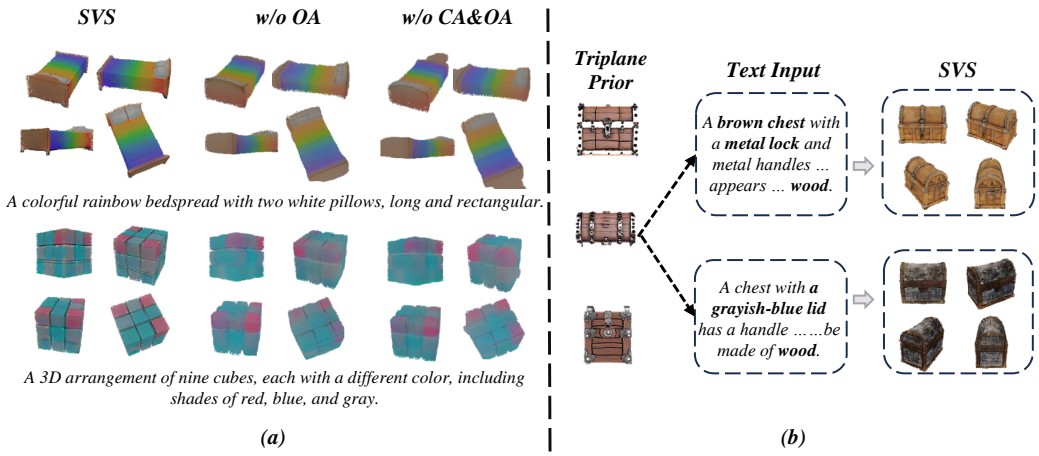

Figure 5: (a) **Ablation Study of Semantic-aligned View Synthesizer (SVS).** Here, cross-attention (CA) and Orthogonalization Attention (OA) aim to improve the quality of view synthesis. (b) **Generalization of SeMv-3D.** When maintaining the same triplane prior, our model can promote the generated objects to be well aligned with different textual semantics, as well as preserve the multi-view consistency.

textures and materials, without changing the main object. From the figure, we observe that based on the same triplane prior, our model can promote the generated objects to be well aligned with different textual semantics, as well as preserve the spatial consistency of the objects across different views. It proves that our method has a strong generalization ability.

## 5 CONCLUSION AND DISCUSSION

In this paper, we study how to effectively and simultaneously achieve semantic and multi-view consistency for the general text-to-3D task. To achieve this target, we propose a SeMv-3D, a novel text-to-3D framework that learns an efficient triplane prior in the TPL to ensure uniformity across all views of an object and align its semantics with the text in the SVS. Noticeably, in SVS, a simple yet effective batch sampling and rendering strategy is proposed that promotes the generation of any view in one single step. Extensive experiments confirm the superiority of our method.

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

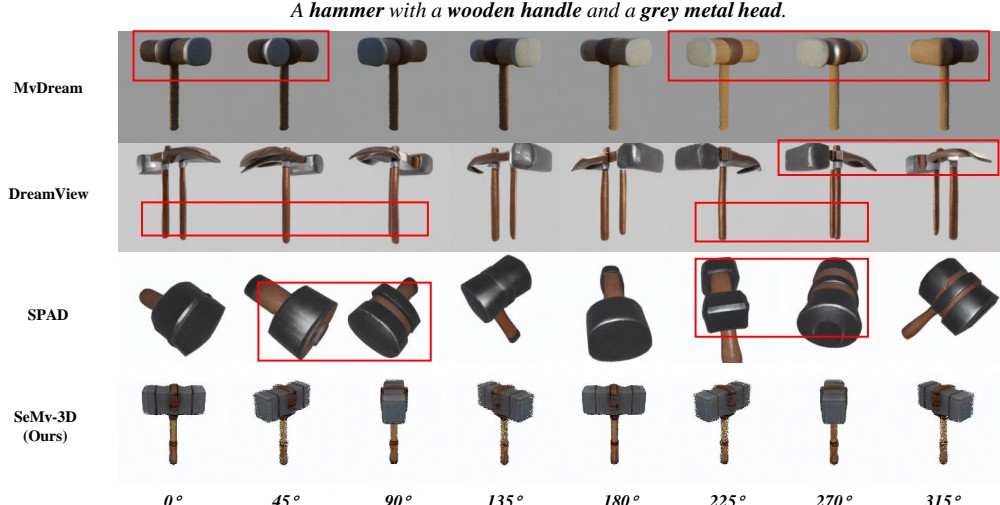

Figure 6: Visual illustration of challenges in Fine-tuning-based Methods.

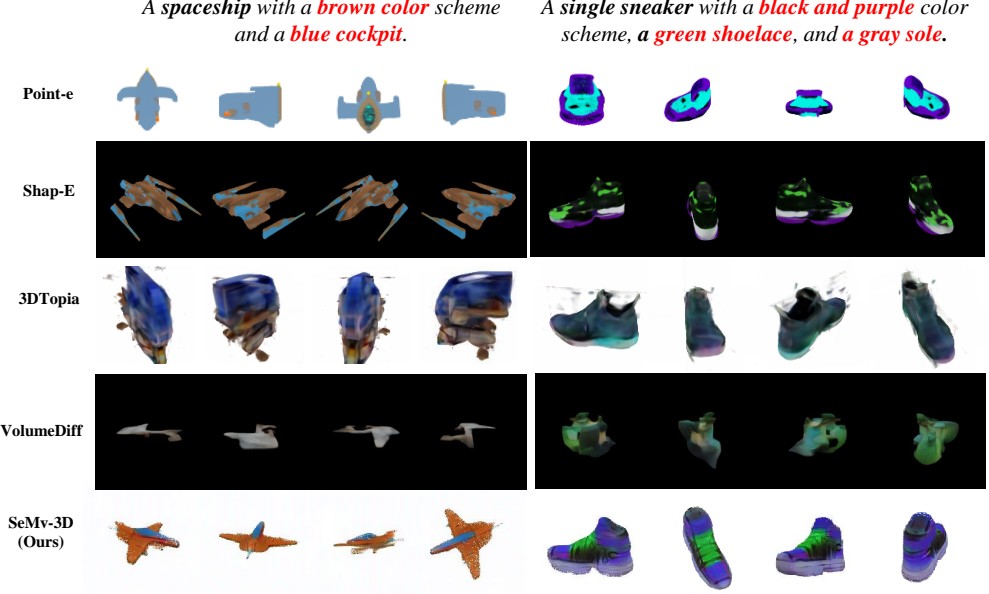

Figure 7: Visual illustration of challenges in Prior-based Methods.

# A APPENDIX

## A.1 DETAILED EXPLANATION OF CHALLENGES IN CURRENT GENERAL TEXT-TO-3D

**Challenges of Fine-tuning-based Methods.** Fine-tuning-based methods can generate high-quality multi-view images; however, they struggle to maintain accurate consistency across different views. Moreover, the number of generated views is often severely limited, restricting the flexibility of these approaches. When we attempt to generate more views beyond the limited number, expanding from four to eight views, the view consistency of MVDream and DreamView almost completely deteriorates, as shown in Fig. 6. Although SPAD makes efforts to achieve arbitrary view generation (as shown in the third row of the figure), and shows some improvement in view consistency without complete discrepancies, it still suffers from significant multi-view inconsistency issues.

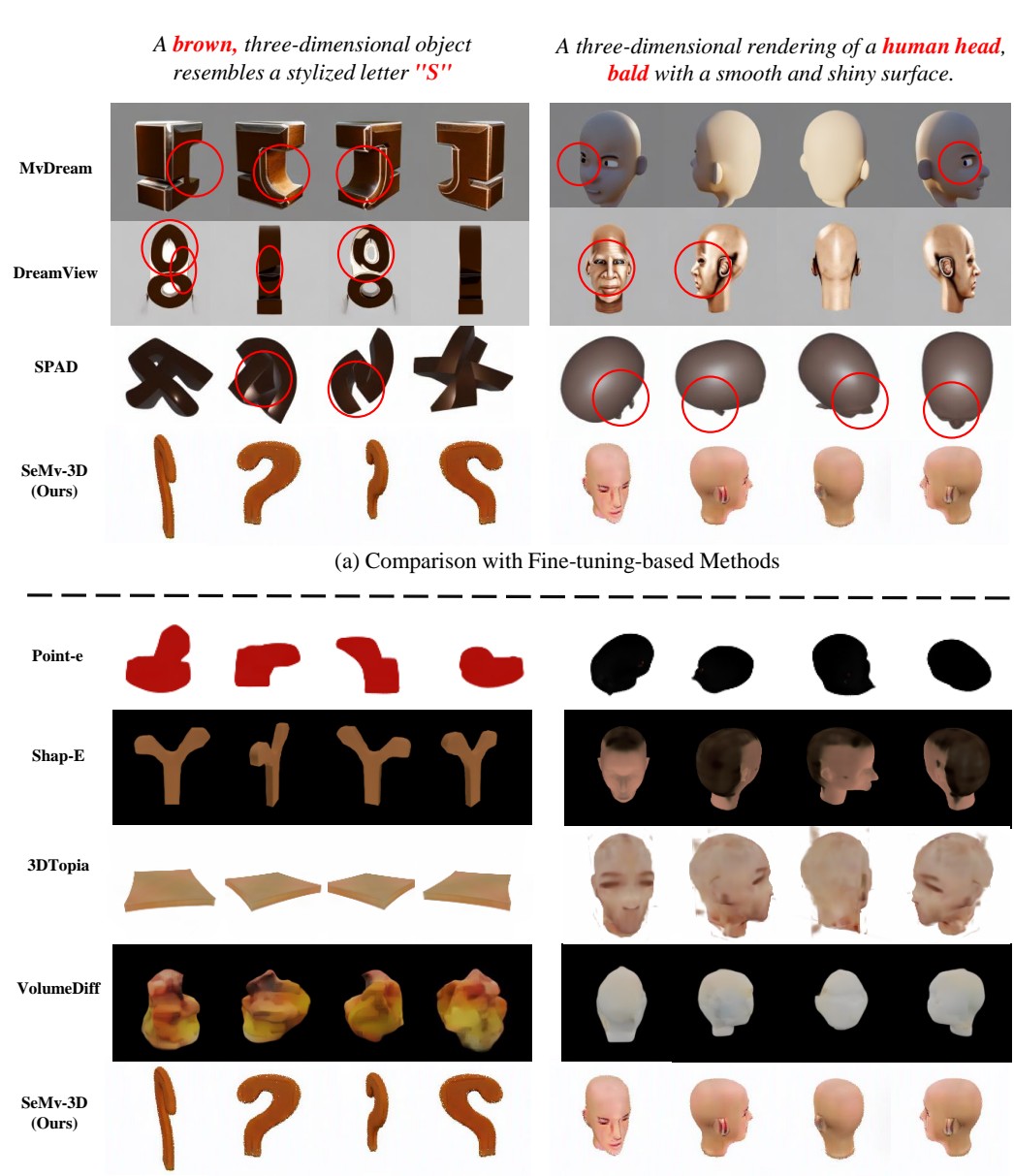

*A **brown,** three-dimensional object resembles a stylized letter "S"*    *A three-dimensional rendering of a **human head**, **bald** with a smooth and shiny surface.*

(a) Comparison with Fine-tuning-based Methods

(b) Comparison with Prior-based Methods

Figure 8: Additional visual comparisons of our SeMv-3D with other General Text-to-3D methods.

**Challenges of Prior-based Methods.** Prior-based methods, while capable of producing relatively consistent multi-view images through 3D rendering techniques, often fail to align the generated 3D content accurately with the input textual semantics. Additionally, the overall quality of the generated 3D content is typically suboptimal. As shown in Fig. 7, even the most advanced method, Shap-e, fails to fully match the semantics of different components in the prompt. For example, the brown scheme and blue cockpit on the left side of the figure cannot be distinguished and are mixed into a brown and blue striped spaceship. Similarly, for green shoelaces, it can only generate a black and green mixed shoe surface. Other methods perform worse, such as Point-e and VolumeDiffusion, which cannot even match the overall color; 3DTopia, on the other hand, only generates a rough outline without details. In summary, both fine-tuning-based methods and prior-based methods have their respective issues. Current general text-to-3D methods cannot achieve both multi-view consistency and semantic consistency simultaneously, which presents the greatest challenge.

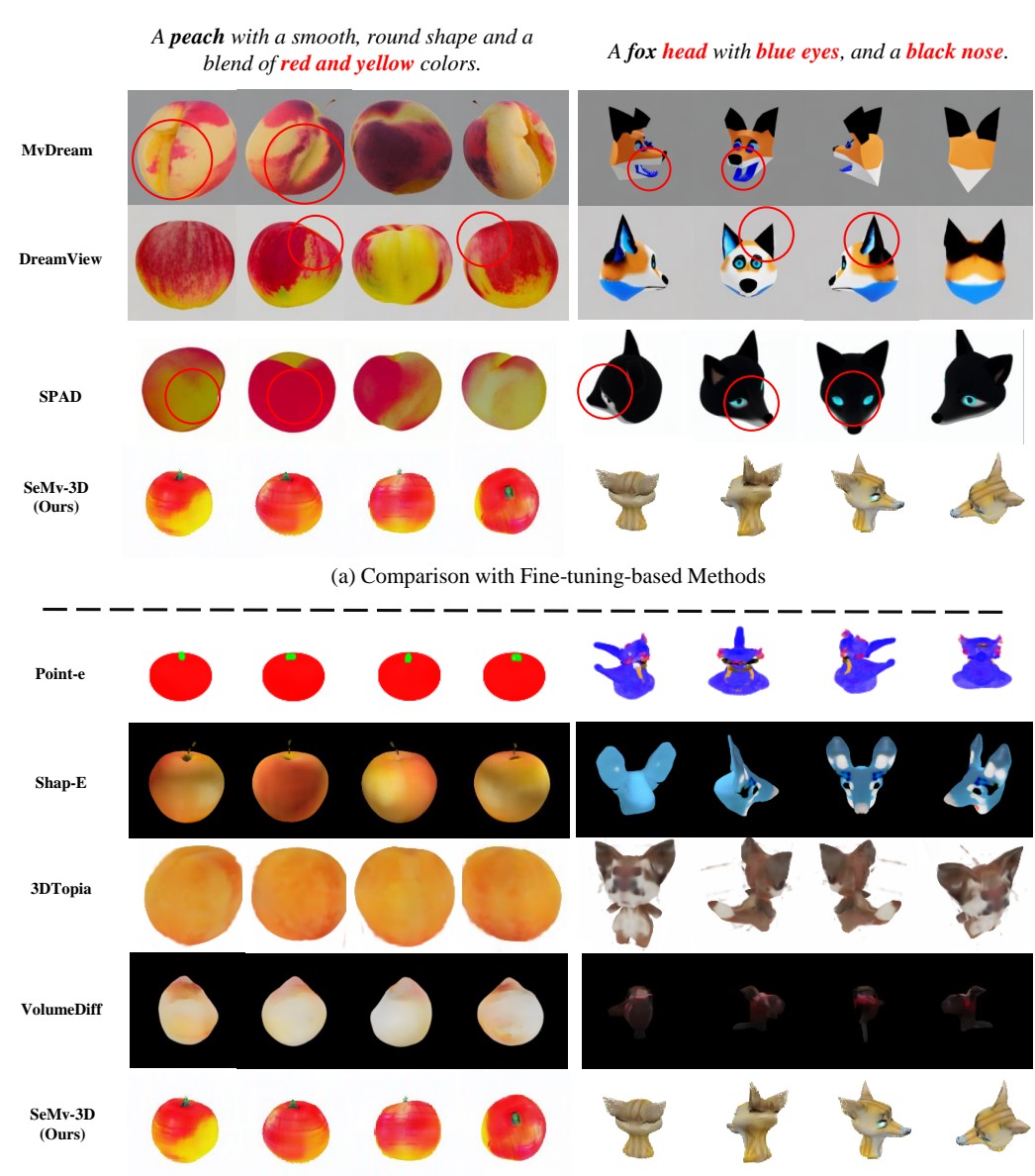

(a) Comparison with Fine-tuning-based Methods

(b) Comparison with Prior-based Methods

Figure 9: Additional visual comparisons of our SeMv-3D with other General Text-to-3D methods.

## A.2 MORE EXPERIMENT RESULTS

### A.2.1 MORE COMPARISON

As shown in Fig. 8 and Fig. 9, we present additional comparative results. Our method demonstrates significantly stronger semantic consistency compared to prior-based methods. For instance, in the case of the fox head with blue eyes and a black nose, some methods fail to generate the head, resulting in either the entire body or no shape at all. Other methods, while generating the head, fail to align fine-grained semantics, such as blue eyes. Methods like Shap-E and Point-E align the blue features with the entire head, unlike our method which aligns the blue semantics precisely with the eyes.

In comparison to fine-tuning-based methods, in addition to the MVDream method discussed earlier, our method shows stronger inter-view consistency and achieves comparable visual outcomes when evaluated against the latest methods such as DreamView and SPAD.

### A.2.2 MORE GENERATION RESULTS

As shown in Fig. 11 and Fig. 12, we present additional visual results to further demonstrate the effectiveness of our method in generating both semantic-aligned and high-fidelity multi-view images.

### A.2.3 QUANTITATIVE RESULTS OF ABLATION STUDY

In the manuscript, we compared the visual results of different modules introduced in TPL and SVS. Here, we supplement these findings with quantitative results as shown in Tab. 2 and Tab. 3 to further validate the effectiveness of these modules.

Table 2: **Quantitative Results of Ablation Study in TPL.** The base model refers to TPL's initialization state, SD-2.1. The symbol '+' indicates the addition of the corresponding module. OR stands for the Object Retention module, TO represents the Triplane Orthogonalization module, and OA signifies the Orthogonalization Attention module. The table shows how each addition affects the Clip Score and Aesthetic Score.

|  | Clip Score | Aesthetic Score |
| --- | --- | --- |
| Base model | **30.99** | **5.38** |
| + OR | 29.31 | 4.14 |
| + OR, + TO | 24.95 | 4.62 |
| + OR, + TO, + OA | 29.67 | 4.28 |

**Ablation Study of TPL.** For the TPL module, the introduction of the OR module allows TPL to generate isolated objects without complex backgrounds while retaining the original object details, as shown in Fig. 4. Although the clip score decreases slightly, it still remains at a high level, dropping by just over one point, demonstrating the effectiveness of the OR module. However, the aesthetic score significantly decreases, likely due to the removal of complex backgrounds.

Next, when the TO module is introduced, Fig. 4 indicates that while the triplanes are learning spatial relationships, the views are inconsistent, and the overall quality of the triplanes drops significantly. The table shows a sharp decline in the clip score with the introduction of the OR and TO modules, but the aesthetic score increases. This supports our hypothesis that vivid colors and complex backgrounds yield higher aesthetic scores.

Finally, with the introduction of the OA module, the quality of the triplanes generated by TPL improves significantly, as illustrated in Fig. 4. The table also reflects that with OA, the clip score is high, second only to the base model. Additionally, the consistency among the triplanes is enhanced, meeting our requirements for high-quality triplane priors. This strongly demonstrates the effectiveness of the OA module.

Table 3: **Quantitative Results of Ablation Study in SVS.** The SVS refers to final version of SVS. The symbol '-' indicates the removal of the corresponding module. OA stands for the Orthogonalization Attention module, CA represents the Cross Attention. The table shows how each removal affects the Clip Score and Aesthetic Score.

|  | Clip Score | Aesthetic Score |
| --- | --- | --- |
| SVS | **31.75** | **4.18** |
| - OA | 29.24 | 3.90 |
| - OA, - CA | 28.81 | 3.92 |

**Ablation Study of SVS.** In the SVS module, we introduced two attentions to align 3D features with semantic representations. As shown in Fig. 5, the addition of the CA module imposes semantic constraints on object generation, even though full alignment with 3D features is not yet achieved.

For instance, artifacts at the foot of the bed are effectively mitigated. As indicated in the table, the inclusion of the CA module improves the CLIP score from 28.81 to 29.24.

Building on this, the subsequent integration of the OA module results in a substantial increase in the CLIP score, rising from 29.24 to 31.75. This demonstrates that self-alignment via the OA module enables more precise matching of semantic and 3D features, further validating the effectiveness of the OA.

### A.3 ILLUSTRATION OF SEMV-3D

**Illustration of TPL.** The core idea of TPL is to fully leverage the knowledge of existing pre-trained models, such as Stable Diffusion (SD), to integrate a 3D-feature-based prior. This approach aims to mitigate the limitations of prior-based methods, including the incompleteness of 2D priors and the potential loss of information during dimensional upscaling. To achieve this, we design two key steps, OR and TO, to transform existing pre-trained models into 3D Triplane prior learners.

Constructing a high-quality and comprehensive Triplane prior requires learning the correspondence among the features of three planes representing the same object in the pre-trained model. However, outputs from pre-trained models often contain complex backgrounds and components unrelated to the prompt, which severely disrupt the learning of spatial correspondences between planes. To address this, we propose the OR module, which removes irrelevant background and focuses on generating the primary content of the prompt.

Building on this, the TO module further learns the spatial correspondences among the three planes. By leveraging the inherent spatial relationships of the Triplane representation, the TO module enables the learning of a comprehensive and integrated 3D prior. This approach significantly improves the quality and consistency of the 3D features, providing a robust foundation for subsequent SVS.

**Illustration of SVS.** The core concept of SVS is to introduce fine-grained semantic matching in the construction of implicit 3D representations. This process fundamentally involves aligning semantic features with orthogonalized 3D features. Unlike traditional prior-based methods, our approach integrates semantic matching into the construction of the triplane implicit field, aiming to achieve precise alignment between semantic features and orthogonalized triplane features.

In practice, instead of merely combining text embeddings with 3D features, our goal is to establish accurate alignment between semantic features and the triplane's orthogonal visual features. However, due to the inherent invisibility of the correspondence between 3D features and specific visual regions, manual alignment of semantic features to visual regions is impractical. To address this challenge, we introduce an orthogonal attention mechanism.

Specifically, we integrate semantic and visual features in the same feature space, allowing them to adaptively align through attention during the implicit triplane reconstruction based on the spatial orthogonal relationships of the triplane. This enables semantic features to automatically align with the most likely visual feature regions, ultimately achieving precise semantic alignment across different 3D visual regions. This approach effectively resolves the challenges of aligning semantics with 3D features and significantly enhances both model performance and generation quality.

### A.4 EXPERIMENTS SETTING

#### A.4.1 IMPLEMENTATION DETAILS.

We train our framework on a subset ($\sim$ 500k objects) of Objaverse dataset (Deitke et al., 2023). We use Stable Diffusion 2.1 to initialize the triplane prior learner (TPL), and train it in the object retention stage for 150k steps with the learning rate $5 \times 10^{-4}$, and in the triplane orthogonalization stage for 60k steps with the learning rate $5 \times 10^{-5}$. The semantic-aligned view synthesizer(SVS) is trained for 100k steps with a learning rate of $5 \times 10^{-4}$. All experiments and training are conducted on eight NVIDIA A6000 GPUs, adopting the AdamW (Loshchilov & Hutter, 2019) optimizer for all stages with $\beta_1 = 0.9$, $\beta_2 = 0.95$, and weight decay 0.03.

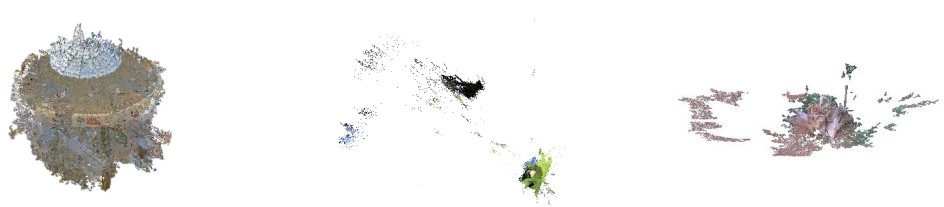

Figure 10: Visualization of some graininess data cases.

### A.4.2 METRICS DETAILS.

**Selection of Metrics.** In open-domain generation tasks, the absence of corresponding ground truth (GT) makes it impractical to use reconstruction metrics such as SSIM (Wang et al., 2004) for evaluation. In image and video generation tasks, specific metrics like FID (Heusel et al., 2017) and FVD (Unterthiner et al., 2018) are commonly designed to comprehensively assess generation quality. However, in the context of 3D generation tasks, no analogous metric (something like F3D) has yet been established.

Therefore, we follow the evaluation protocol used in prior works. Objectively, we assess generation quality and semantic alignment using the CLIP score. Subjectively, we conduct a user study to evaluate multi-view consistency and generation quality comprehensively. Additionally, to further demonstrate the high quality of the generated results from multiple perspectives, we incorporate the aesthetic score as a supplementary evaluation metric.

**User Study Setting.** Due to the lack of diverse objective evaluation metrics for general text-to-3D methods, user studies are commonly employed to further validate the effectiveness of these approaches. In this experiment, we invited 40 highly educated individuals with undergraduate degrees or higher to participate in the evaluation. Among them, approximately 20 have experience in AI-related research or work, 10 are engaged in artistic professions, and the remaining 10 are involved in fields such as civil engineering, architecture, and sports.

In practice, each user is first given 9 groups of generated four views (e.g., $0°, 90°, 180°, 270°$) and the corresponding prompts. Then, they are asked to select their preferred method from three levels, including Users Prefer, Semantic Consistency, and Multi-View Consistency. They first evaluate the overall quality and selected their preferred option (**Users Prefer**). Then, based on consistency, they separately identify the method with the highest semantic alignment (**Semantic Consistency**) and the method with the greatest consistency across different views (**Multi-View Consistency**).

### A.5 LIMITATION.

**Lack of High-Quality Dataset.** The goal of the general text-to-3D task is to learn a generic model that can generate various objects in a feed-forward manner. However, this field lacks high-quality large-scale text-3d pairing data, influencing the quality of generation.

**Graininess Issues.** We observe that fragmented white graininess occasionally appears in certain thin, sheet-like objects and along the edges of some objects. Upon analysis, we identify two potential factors: First, the dataset contains broken objects that are not fully filtered out, and their characteristics (as illustrated in Fig. 10) are learned by the model, resulting in discrete noise artifacts during generation. Second, due to computational resource limitations, the model is trained with a relatively small batch size, which may have impacted its robustness, particularly in handling thin objects.

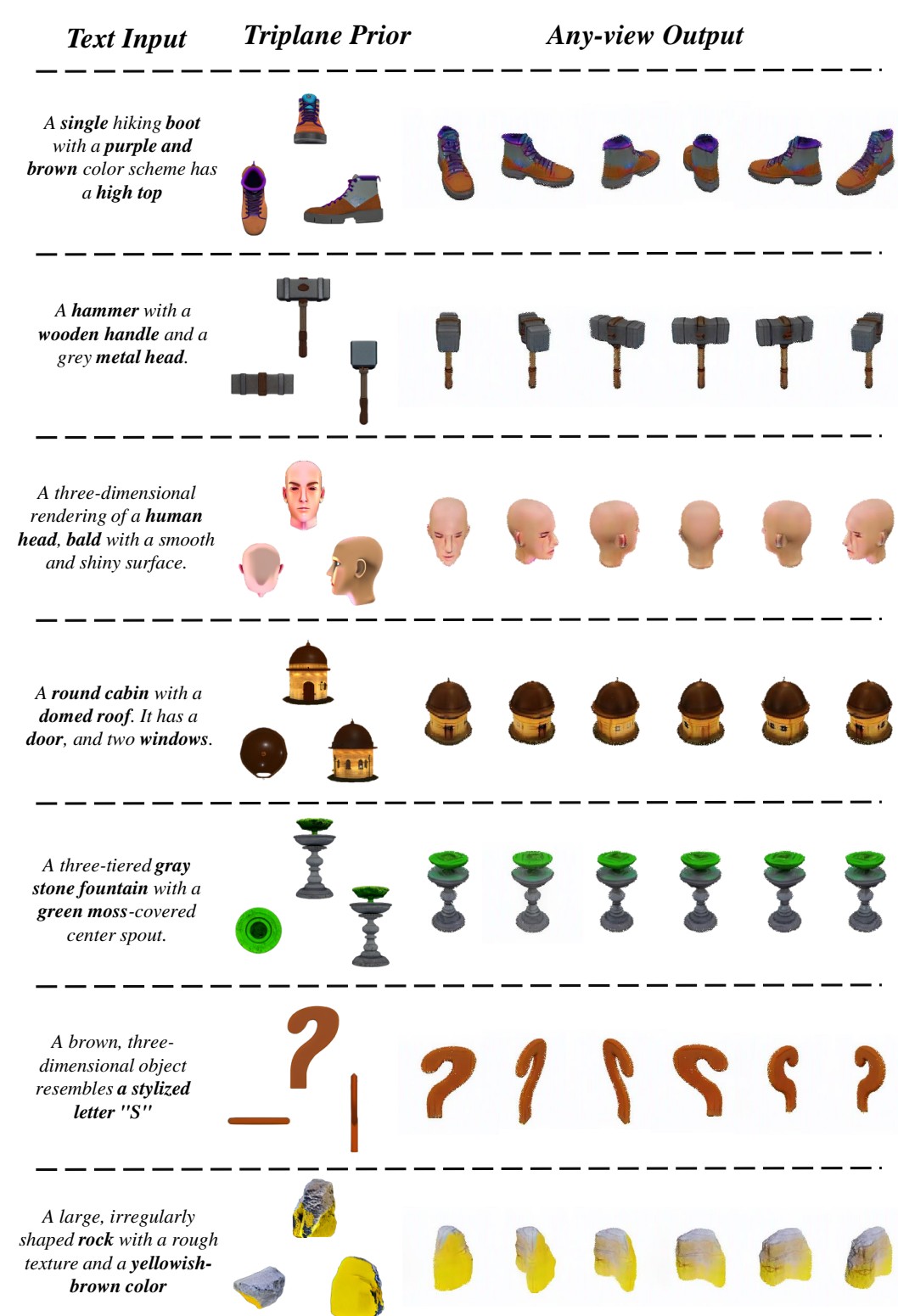

Figure 11: Additional triplane visualization and results of our SeMv-3D on Text-to-3D task.

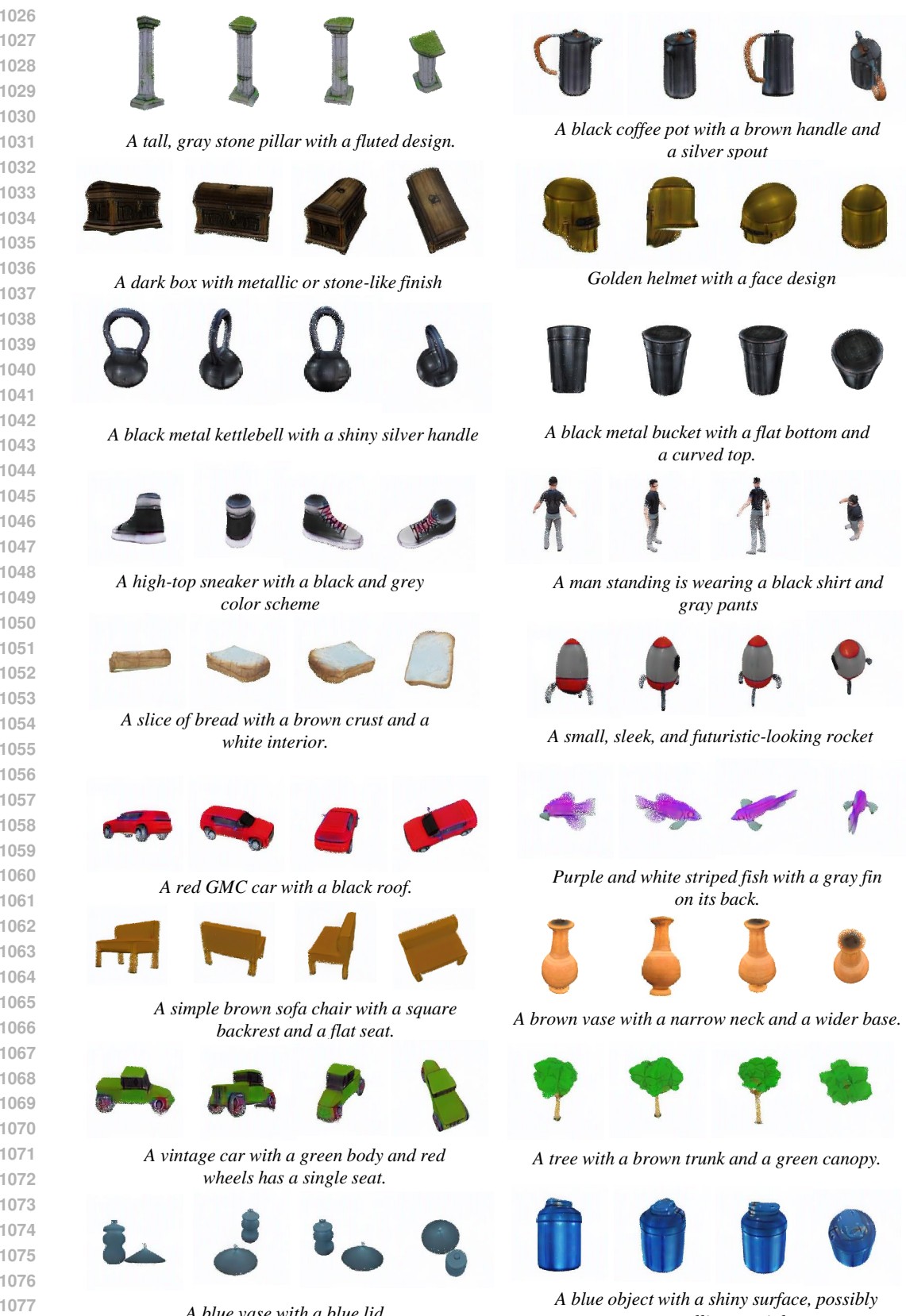

Figure 12: More visual results of our SeMv-3D on Text-to-3D task.

