# OpenReview forum: "SeMv-3D: Towards Semantic and Mutil-view Consistency simultaneously for General Text-to-3D Generation with Triplane Priors"
_ICLR.cc/2025/Conference — Submitted to ICLR 2025_

### Official Review · Reviewer_mXqv · 2024-10-21

**Soundness:** 3
**Presentation:** 3
**Contribution:** 3
**Rating:** 6
**Confidence:** 4

**Summary:**

This paper proposes a novel framework, SeMv-3D, to ensure both semantic and multi-view consistency in generating 3D representations from text. It introduces a TPL module for learning the triplane prior and an SVS module for aligning textual and 3D visual information.

**Strengths:**

- This paper is well-organized and well-motivated, targeting the text-to-3D problem with promising application prospects.
- This paper effectively summarizes the limitations of existing methods in terms of semantic and multi-view consistency.
- This paper makes a commendable attempt to address these issues within a unified framework.

**Weaknesses:**

- The compared baseline methods are limited, lacking comparisons with recent methods such as [1] and [2].
- The quality of the generated 3D models is limited, with granular and discontinuous surfaces visible in rendered images, particularly in Figures 3, 6 and 7.

[1] DreamView: Injecting View-specific Text Guidance into Text-to-3D Generation, ECCV 2024.
[2] DiverseDream: Diverse Text-to-3D Synthesis with Augmented Text Embedding, ECCV 2024.

**Questions:**

- Many rendered images have noticeable graininess, such as the result on the right side of Figure 3 and the fish in Figure 6. Can you analyze the reason?

---

> ### Author Response · Authors · 2024-11-28
>
> Thank you for your recognition of our method and paper. Below, we address the questions and weaknesses you identified:
>
>
>
> 1. ### **Weaknesses**
>
>    1. **Comparison with Existing Methods**
>       We sincerely appreciate your suggestions for additional comparison methods. In response, we have supplemented the relevant comparative results on page 8 and in Appendix A.2.1. Based on both quantitative and qualitative evaluations, we observe that while DreamView achieves higher semantic consistency compared to MVDream, its multi-view consistency deteriorates. Additionally, DreamView does not address the limitation of generating a restricted number of views, which SPAD partially resolves. Consequently, in the user study, DreamView's preferred rate was lower than that of MVDream.
>
>       In contrast, our **semv3d** method surpasses DreamView in multi-view consistency while achieving comparable levels of semantic consistency and generation quality. Furthermore, our approach enables **any-view multi-view generation**, providing a distinct advantage. Thus, we believe that **semv3d** demonstrates overall superior performance.
>
>       Additionally, it is worth mentioning that the second method, **DiverseDream**, being an SDS optimization-based method, is not a feed-forward approach and was therefore excluded from our comparison.
>
>    2. **Specific Weakness Clarification**
>       Please refer to the response to your question below for further elaboration on the identified issue.
>
>    ------
>
>    ### **Questions**
>
>    Regarding the appearance of minor scatter points in the generated outputs, we identify two potential causes:
>
>    1. **Dataset Quality**: The dataset contains broken objects that were not fully filtered out during preprocessing. As shown in Figure 10 of Appendix A.5, the model learns the characteristics of these fragmented objects, which manifest as discrete noise artifacts during generation.
>    2. **Batch Size Limitation**: Due to computational resource constraints, the model was trained with a relatively small batch size. This limitation may have impacted the model's robustness, particularly in handling thin or delicate object structures.
>
>    We acknowledge these issues and aim to address them in future work by improving dataset preprocessing and exploring methods to enhance model robustness under resource constraints.

---

> > ### Comment · Reviewer_mXqv · 2024-12-02
> >
> > Thanks for your feedback! My concerns are addressed and I decide to keep my rating unchanged.

---

### Official Review · Reviewer_2wmw · 2024-10-31

**Soundness:** 3
**Presentation:** 3
**Contribution:** 2
**Rating:** 5
**Confidence:** 5

**Summary:**

This paper addresses the generalized text-to-3D generation problem. It proposes SEMV-3D -- a text-to-3D generative model composed of two stages: The first stage converts an input text prompt into three orthogonal images forming an “RGB-triplane,” while the second stage maps this “RGB-triplane” to a true radiance field triplane. Specifically, SEMV-3D introduces a Triplane Prior Learner (TPL) module in the first stage, which fine-tunes Stable Diffusion to generate a background-free front view, then refines additional layers (front, side, and top views) to produce the final RGB-triplane. In the second stage, the Semantic-aligned View Synthesizer (SVS) transforms the RGB-triplane into DINO tokens, which are concatenated with semantic descriptors of different prompt parts and passed through a transformer to create the true radiance field. Experiments are conducted in several settings including quantitative, qualitative, and user-study results to demonstrate the effectiveness of the proposed method.

**Strengths:**

+ The paper is well-organized and written.
+ The introduced modules, particularly in the TPL stages, appear effective; notably, the Orthogonalization Attention (OA) enhances view consistency and 3D detail, likely contributing significantly to the final 3D model.
+ The overall pipeline is logically structured.

**Weaknesses:**

+ The lack of quantitative results in the ablation study makes it challenging to validate the effectiveness of individual modules.
+ The reasoning behind dividing the TPL stage into two substages, Object Retention (OR) and Triplane Orthogonalization (TO), is unclear.
+ Equation 6 lacks clarity: the meaning of ￼ is undefined, and the OA module should have two input arguments, though equation 6 shows only one.
+ It is unclear why a widely recognized metric like FID (or its variants) is not used to evaluate the model’s generative performance.
+ The method for computing semantic information in the SVS stage is not well-explained.
+ Line 232 should reference Fig. 2 (c) for easier comprehension.
+ There is no time comparison with MVDream or similar approaches.
+ Figures 4 and 5 show results of lower quality compared to MVDream.
+ Visualizations and examples of the RGB-Triplane are needed, as it is an uncommon term.

**Questions:**

+ In the TPL stage, could merging the two substages impact performance? Since the TO modules seem to encompass the OR modules, could it be possible to train the OR modules individually for each orientation in the first substage, and then aggregate the trained weights with the OA modules in the second? Additional insights on this would be valuable.
+ Could you provide more detail on Equation 6 and the computation of semantic information in the SVS stage?
+ Addressing the choice of metrics would clarify the evaluation approach and strengthen the paper.
+ A comparison between Orthogonal Attention and Epipolar Attention from [a] and [b] would be beneficial.

[a] Kant, Yash, et al. "SPAD: Spatially Aware Multi-View Diffusers." Proceedings of the IEEE/CVF Conference on Computer Vision and Pattern Recognition. 2024.

[b] Huang, Zehuan, et al. "Epidiff: Enhancing multi-view synthesis via localized epipolar-constrained diffusion." Proceedings of the IEEE/CVF Conference on Computer Vision and Pattern Recognition. 2024.

---

> ### Author Response · Authors · 2024-11-28
> **Response to weakness**
>
> Thank you for your valuable suggestions. We have updated the manuscript to address several details that were previously underexplained. Below, we respond to the weaknesses you raised:
>
>
>
> **Weakness：**
>
> 1. Quantitative Results: We acknowledge your point regarding quantitative results. To address this, we have supplemented the corresponding quantitative analyses in the supplementary materials, specifically in Appendix A.2.3.
>
> 2. Explanation of TPL Stages: We acknowledge the previous lack of clarity regarding the two-stage structure of TPL. The manuscript has been revised, and a detailed explanation will be provided in response to your first question below.
>
> 3. We apologize for the omission in the definition of the Transformer in Equation 6. To facilitate understanding, we have reorganized the logic and updated the manuscript 3.3.1 in page 5. An explanation is also provided in response to question 2. The OA module does indeed have a single input, which we will clarify in the 3.3.1.
>
> 4. FID and similar metrics require ground truth data for feature similarity calculations, which is not applicable to text-to-3D tasks lacking ground truth.  We use alternative metrics, as explained in response to question 3, and have elaborated on the rationale in Appendix A.4.2.
> 5. SVS Module Explanation: We regret the lack of clarity in the initial explanation of the SVS module. The manuscript has been updated to better describe its motivation and implementation. Please refer to our response to your question 2 for further details.
>
> 6. Thank you for your suggestion. We have updated the manuscript to clarify references to the OA module where appropriate.
>
> 7.  Inference Time Comparison:
>
> |     Methods     | Inference Time |
> | :-------------: | :------------: |
> |     MVDream     |       6s       |
> |    DreamView    |       6s       |
> |      SPAD       |      66s       |
> |     Point-e     |      13s       |
> |     Shap-e      |      65s       |
> | VolumeDiffusion |      58s       |
> |     3Dtopia     |      27s       |
> |     SeMv-3D     |      46s       |
>
>  While our method does not achieve the fastest inference time, it demonstrates strong overall performance. Compared to the state-of-the-art prior-based method Shap-e, our method achieves a **29% speed improvement**. Notably, methods like MVDream and DreamView, which are the fastest, are limited to generating only four views, whereas our approach supports **arbitrary view generation**. Additionally, compared to SPAD, an any-view method with an inference time of **66 seconds**, our method achieves a **~30% speed improvement** while exhibiting superior multi-view consistency.
>
> 8. Explanation of Figures 4 and 5: The results presented in Figure 4 are merely visualizations of the Triplane priors and do not represent the final outputs. Nevertheless, at this stage, the visual quality of the Triplane results is already comparable to those of methods like MVDream, even if the visual appeal may seem less striking, likely due to differences in style.
>
>    Further analysis of Figure 5 reveals that, due to the inherent randomness in Triplane generation, we adopted orthogonal views (frontal, top-down, and side) from real data as the Triplane inputs to ensure consistency across different SVS configurations. Since the data itself leans more toward a realistic style, our results may appear less visually striking compared to the more artistic style of images generated by methods like MVDream.
>
> 9. We regret that the generated results do not include the corresponding Triplane priors. Due to the inherent randomness in the model's generation process, each Triplane is different for every run. To address this, we have supplemented additional results incorporating Triplane priors in Appendix A.2.2 and provided detailed comparative results in Figure 9 of Appendix A.2.1. These supplementary materials aim to offer a more comprehensive understanding and validation of our method's performance.

---

> > ### Author Response · Authors · 2024-11-28
> > **Response to Questions**
> >
> > Thank you for your valuable suggestions. We have updated the manuscript to address several details that were previously underexplained. Below, we respond to the questions you raised:
> >
> > **Question:**
> >
> > 1. Apologies if I did not fully understand your question earlier. If your inquiry is about whether the TO module and OR module can be trained separately, followed by introducing the OA module in a later stage, our perspective is as follows: The two sub-stages of the TPL phase (OR and TO) are not entirely independent or isolated. The OR module plays a crucial role in preserving the primary content while removing irrelevant background, whereas the TO module aims to learn the spatial correspondence among the three planes. If the TO module were trained directly on results generated by SD, the complex background produced by SD would severely disrupt the learning of spatial relationships. Therefore, we opted to train the TO module after the OR stage, ensuring that irrelevant backgrounds are removed, allowing the model to focus on spatial alignment of the main content. Regarding the OA module, while it might theoretically be introduced in a later stage, our ablation studies demonstrate that training the three-plane spatial correspondence without OA leads to suboptimal results. Rather than learning a subpar spatial correspondence and then attempting to improve it by introducing the OA module later, it is more effective to incorporate the OA module from the beginning. This approach allows the model to directly learn superior spatial correspondences, as the OA module is specifically designed to enhance the spatial alignment of Triplane features and is therefore indispensable during training.
> >
> >
> >
> > 2. The implementation details of this semantic matching mechanism are illustrated in Equation 6 in page 6.
> >
> > First, we extract the triplane visual features, denoted as $token_{tri}$. These features are then enriched with semantics $T$ through CA, represented as $CA(token_{tri}, T)$. As shown in the figure 5, using $token_{tri}$ directly for reconstruction results in surface blurriness, lack of detailed textures, and significant edge artifacts, such as at the foot of the bed on the left side of the figure. Introducing CA mitigates these edge artifacts due to semantic constraints, although texture details are still lacking.
> >
> > Then, we apply OA to achieve self-spatial orthogonal alignment, further matching semantic information with orthogonal features. As illustrated in the leftmost column of the figure, after incorporating OA on top of CA, details such as the pillows on the bed are fully generated.
> >
> >
> >
> > 3. Evaluation Metrics: You are correct that text-to-3D tasks require metrics that evaluate semantic and 3D alignment alongside multi-view consistency. To this end, we use the **CLIP score** to measure semantic consistency and **User study** to evaluate the quality of the generated objects. We also consider the number of views generated per inference step as a criterion for generation efficiency. These metrics are detailed in Appendix A.4.2.
> >
> >
> >
> > 4. We sincerely appreciate your suggestions for additional comparison methods. The SPAD method, leveraging the EA strategy and building on MVDream, successfully achieves Any-view multi-view generation, similar to our approach. This addresses the limitation of existing fine-tune-based methods, which can only generate a limited set of views. In response, we have updated our main experiments on page 8. From both quantitative and qualitative results, it is evident that while SPAD achieves arbitrary view generation, its output quality is significantly lower than MVDream, and issues with view consistency remain unresolved. Similarly, additional comparisons provided in Appendix A.2.1 further substantiate this conclusion. Therefore, we believe our semv3d outperforms SPAD in overall performance. Regarding the other EA-based method, EpiDiff, its input is image-based, making it a 2D-to-3D generation task. As this fundamentally differs from our text-to-3D generation task, we did not include it in the comparison.

---

### Official Review · Reviewer_KT6v · 2024-11-01

**Soundness:** 3
**Presentation:** 3
**Contribution:** 3
**Rating:** 6
**Confidence:** 4

**Summary:**

This paper presents SeMv-3D, a framework for text-to-3D generation that aims to address two main challenges in the field: semantic consistency and multi-view consistency. SeMv-3D utilizes a Triplane Prior Learner (TPL) and a Semantic-aligned View Synthesizer (SVS) to achieve dual consistency. TPL captures spatial coherence across views using a triplane representation, while SVS aligns three-dimensional features with textual semantics. This framework utilizes a batch sampling and rendering method that allows for generating arbitrary views in one feed-forward step. SeMv-3D outperforms existing methods in extensive experiments, achieving state-of-the-art performance despite limitations in text-to-3D data and computational resources.

**Strengths:**

1. SeMv-3D effectively addresses challenges in achieving both multi-view and semantic consistency in text-to-3D generation.

2. The TPL and SVS modules enable SeMv-3D to maintain high visual-textual alignment while ensuring 3D consistency across different views.

3. The batch sampling and rendering approach enables efficient and flexible view generation in a single inference pass.

4. Extensive experiments demonstrate SeMv-3D’s qualitative and quantitative improvements over state-of-the-art methods.

**Weaknesses:**

1. The lack of high-quality, large-scale text-3D paired data limits the model's ability to optimize for generalizability.

2. Limited resources impact convergence, which can influence the stability and quality of the model's performance in complex scenarios.

3. Clarify the novelty of the proposed semantic-aligned method. What is the semantic-aligned strategy, and how can it be applied?

**Questions:**

This paper claims to use semantic and view-consistent methods to achieve high-quality text-to-3D. However, it does not clarify the semantic consistency process. Please provide more details to support the contributions.

1. How does SVS achieve semantic alignment?

2. Details on SVS’s mechanisms to match text with 3D features in latent space would support the semantic consistency claim?

3. Clarify if TPL’s spatial correspondences enhance semantic alignment across views?

4. Explain if TPL features are iteratively refined to improve SVS alignment?

5. Specify how simultaneous multi-view generation strengthens semantic coherence?

6. Evidence comparing SeMv-3D’s semantic consistency to other methods would substantiate this contribution?

---

> ### Author Response · Authors · 2024-11-28
>
> Thank you for your valuable suggestions. We have updated the manuscript to polish some of the details not yet fully explained previously. Here, I will first address the questions you raised:
>
>
>
> **Weakness**:
>
> 1. Dataset Quality and Scale: Large-scale, high-quality datasets can significantly enhance model performance. To ensure the acquisition of a sufficiently high-quality, large-scale dataset, we have implemented the following measures:
>
>   \- We selected Objaverse, the largest open-source 3D dataset available at the time.
>
>   \- Given the varying quality of the dataset, we conducted necessary data preprocessing before training to filter out low-quality data.
>
>
>
> 2.  Limitations of Computational Resources: Limited computational resources can indeed affect model convergence. To maximize model performance, we adopted the following strategies during training:
>
>   \- Increased batch size and utilized gradient accumulation to further enhance robustness.
>
>   \- Randomly sampled generated views during training to improve generalization.
>
>   \- Applied a dynamic cosine learning rate adjustment strategy based on dataset scale to facilitate better model convergence.
>
>
>
> 3. Innovation in Semantic Matching Strategy: The core innovations and implementation of our semantic matching strategy are detailed in the response to the subsequent questions.
>
>
>
> **Questions**:
>
> 1.  The core idea behind semantic matching in SVS is to align semantic features with orthogonalized 3D visual regions. Unlike prior-based methods, which do not incorporate semantic matching during the formation of implicit fields, our approach introduces semantic matching during the construction of the triplane implicit field. During the implicit triplane reconstruction, which leverages the spatial orthogonal relationships of the triplane, attention mechanisms guide the adaptive alignment between semantic and visual features. This process allows semantic features to naturally align with the most likely corresponding visual feature regions, ultimately ensuring precise semantic alignment across various 3D visual regions.  More illustration Information is provided in Appendix A.3.
>
>
>
> 2.  The implementation details of this semantic matching mechanism are illustrated in Equation 6 in page 6.
>
>    First, we extract the triplane visual features, denoted as $token_{tri}$. These features are then enriched with semantics $T$ through CA, represented as $CA(token_{tri}, T)$. As shown in the figure 5, using $token_{tri}$ directly for reconstruction results in surface blurriness, lack of detailed textures, and significant edge artifacts, such as at the foot of the bed on the left side of the figure. Introducing CA mitigates these edge artifacts due to semantic constraints, although texture details are still lacking.
>
>    Then, we apply OA to achieve self-spatial orthogonal alignment, further matching semantic information with orthogonal features. As illustrated in the leftmost column of the figure, after incorporating OA on top of CA, details such as the pillows on the bed are fully generated.
>
> 3. Our TPL aims to generate high-quality triplane priors by learning the spatial correspondence among the three planes. This primarily ensures view consistency while also enhancing the semantic consistency of each plane to some extent. As illustrated in the figure 4 of page 10, the introduction of OA not only significantly improves view consistency but also maintains semantic consistency. For instance, the red and gray shoes demonstrate both view and semantic consistency after incorporating OA. In contrast, the method on the left, which does not employ OA, maintains view consistency but loses substantial semantic information.
>
> 4. Our method is a feed-forward approach, without any iterative optimization of features throughout the process. Therefore, the output of TPL is directly fed into SVS.
>
> 5. Our SVS generates an arbitrary number of multi-view images through ray sampling and rendering from the triplane. Essentially, we enhance the semantic consistency of the multi-view images by improving the semantic consistency of the triplane implicit field, achieved through the functionality outlined in response 1/2.
>
> 6. In our main experiment, we have already demonstrated the comparative results of semantic consistency. For instance, in Figure 3 of page 8, the green shoes meet both the high-top design and single shoe constraint, which only our SeMv-3D can achieve. However, as you pointed out, the experimental results are not yet sufficient. Therefore, to more comprehensively illustrate our semantic consistency, we have included additional experimental results in the figure 7 of Appendix A.1 and figure 8/9 of Appendix A.2.1.

---

> > ### Comment · Reviewer_KT6v · 2024-11-29
> >
> > I have read the rebuttal carefully. My concerns have been addressed, so I increased the score.

---

### Official Review · Reviewer_VfgQ · 2024-11-04

**Soundness:** 2
**Presentation:** 2
**Contribution:** 2
**Rating:** 5
**Confidence:** 4

**Summary:**

This paper presents a feed-forward text-to-3D generation framework, which consists of a triplane prior learner model and a semantic-aligned view synthesizer. Experiments are conducted to validate the effectiveness of the proposal.

**Strengths:**

S1. The proposed triplane prior learner is reasonable.

S2. Learning a high-quality feed-ward text-to-3D generation model is a promising research direction.

S3. The proposed orthogonal attention is suitable for the triplane network.

**Weaknesses:**

W1. What is the evaluation set used by authors? I am wondering whether the authors use standard benchmarks to evaluate their methods.

W2. The authors only show two or three qualitative comparison examples in Figure 3. It is encouraged to provide more comparison examples to avoid cherry-picking results.

W3. The authors only conduct qualitative ablation studies in Sec 4.5. Quantitative ablation studies are also necessary to better validate the effectiveness of each design.

W4. What is the inference time of the proposed method?

**Questions:**

Please see the Weaknesses.

---

> ### Author Response · Authors · 2024-11-28
>
> Thank you very much for your valuable and constructive feedback. Below, we address each of your points in detail:
>
>
>
> 1.  Benchmark for General Text-to-3D: Due to the lack of a standard benchmark for general text-to-3D tasks, we employed GPT to automatically generate prompts, ensuring a fair and consistent comparison across different methods. In the future, we will publicly release our test prompt set to facilitate result reproduction and further validation by the community.
>
>
>
> 2. Comparative Results: As you rightly pointed out, it was an oversight to include only our method’s results without comparative analysis in the appendix. We have now addressed this by **adding more comparative results** in Appendix A.2.1 for a comprehensive evaluation.
>
>
>
> 3. Quantitative Ablation Studies: While the visual results provide strong evidence for our method's effectiveness, we agree that **adding quantitative ablation studies would strengthen our argument**. Following your suggestion, we have included the corresponding tables in Appendix A.2.3 to provide a more thorough analysis.
>
>
>
>
>
> 4. Our SeMv-3D method achieves an average inference time of approximately **46 seconds** per prompt, representing a **~29% speedup** compared to the current state-of-the-art **prior-based model, Shap-E**, which requires **65 seconds**. Furthermore, our method demonstrates superior semantic consistency and enables any-view generation. In comparison to the previous any-view method, SPAD, which has an average inference time of **66 seconds**, our method achieves a **~30% speedup** while also exhibiting better multi-view consistency. These results highlight that SeMv-3D not only offers significant advantages in semantic and multi-view consistency but also achieves substantial improvements in efficiency.

---

> ### Comment · Reviewer_VfgQ · 2024-12-02
>
> I appreciate the author's clarification in their rebuttal. I tend to retain my original rating, as the evaluation set used in this paper is not convincing for me. The authors need to evaluate their methods on a standard benchmark, such as T3Bench.

---

### Official Review · Reviewer_b6y7 · 2024-11-05

**Soundness:** 3
**Presentation:** 2
**Contribution:** 2
**Rating:** 3
**Confidence:** 5

**Summary:**

This work proposes a feedforward text-to-3D generation framework with multi-view consistency and semantic alignment. Specifically, it fine-tunes a well-trained diffusion model to produce triplane priors. To achieve better semantic alignment, it further introduces the DINO feature in the semantic-aligned view synthesizer module.

**Strengths:**

1. This work proposes a triplane-based pipeline to keep multi-view consistency and semantic alignment.
2. The experiments showcase better consistency than MVDream.

**Weaknesses:**

1. The motivation behind this work appears insufficient.
  - The choice of MVDream-like methods for comparison seems misplaced. Fine-tuned diffusion models typically require additional optimization steps like NeRF or NeuS reconstruction to generate 3D shapes directly. A more appropriate comparison could be made with works such as "ET3D: Efficient Text-to-3D Generation via Multi-View Distillation," which aligns closely with the task at hand.
  - Concerns arise regarding semantic consistency. While prior-based methods have struggled with semantic alignment issues stemming from overfitting to 3D datasets, SEMV-3D's Semantic View Synthesis (SVS) mechanism appears to maintain semantic alignment, largely relying on DINO features. Similar strategies have been observed in instant3D and GRM.

2. The paper overlooks multiview reconstruction-based feedforward pipelines, such as instant3D, LGM, and GRM, which excel in semantic consistency. Notably, instant3D, incorporating DINO and Triplane, shares striking similarities with this approach.
- The claim of Triplane's 'efficiency' in Line 159 may not be entirely justified, particularly compared with 3D Gaussians.

**Questions:**

- What is the inference time of SEMV-3D per prompt?

- Figure 5 lacks a text prompt. The CA exhibits less benefit in Figure 5.

---

> ### Author Response · Authors · 2024-11-28
>
> Thank you for recognizing our paper and method, as well as for providing valuable suggestions. In this response, we will first address the weaknesses you highlighted and then provide detailed answers to your questions.
>
> **Weakness**
>
> 1. Regarding the motivation, existing general text-to-3D tasks that support **direct inference in a feed-forward manner** can be broadly categorized into two types. Prior-based methods, such as Shap-E, excel in view consistency but suffer from **poor semantic alignment and subpar generation quality**. Fine-tuning-based methods, like MVDream, achieve **high semantic consistency and superior quality** but are limited by **view inconsistencies** and the ability to generate only a few views. Our motivation is to propose a method that combines the strengths of both approaches, achieving **arbitrary view generation** with **high multi-view consistency and semantic consistency**, while also delivering refined and high-quality results. This forms the foundation of our approach.
>
>    - MVDream primarily relies on fine-tuning diffusion-based models, such as text-to-image or multi-view generation models, without incorporating 3D constraints like NeRF used in Shap-E. Instead, it leverages the diffusion model's capability to directly generate multi-view images, which we classify as **fine-tuning-based methods**.
>
>      In contrast, ET3D belongs to the category of **optimization-based methods**, which involve iterative optimization of 3D representations like NeRF. For example, ET3D employs a distillation-based strategy to iteratively refine 3D structures. This fundamental difference from feed-forward general text-to-3D tasks is why ET3D and similar approaches are excluded from our comparative analysis.
>
>    - In our approach, DINO is used solely as a tool for **extracting visual features**. We chose DINO features due to their ability to preserve rich visual characteristics, which is likely why methods such as Instant3D and GRM also employ them. However, as illustrated in Figure 5, solely relying on features extracted by DINO is insufficient to achieve optimal results.
>
>      Our primary focus is on **designing a novel semantic alignment mechanism based on the Triplane features themselves**. The detailed principles and implementation of this mechanism can be found in the Sec.3.3.1 and Appendix A.3 of the paper. By integrating visual features with this mechanism, we further enhance the precision and effectiveness of semantic alignment, leading to superior overall performance.
>
> 2. Multi-view-based reconstruction tasks essentially initialize with a multi-view setup based on general text-to-3D models like MVDream but focus on image-based 3D reconstruction. They don't align with our general text-to-3D baseline model, although future work on extending SeMv-3D to 3D reconstruction could consider comparisons with these methods.
>    - while we have not reasonably compared the efficiency of triplane with new representations like 3DGS, further exploration of 3DGS's efficiency in generalization tasks could be valuable. Nonetheless, the triplane representation significantly reduces computational complexity compared to traditional 3D representations, meeting the needs of our generalization tasks while retaining excellent expressive capability. This efficiency justifies our claim.
>
>
>
> **Questions**:
>
> 1. Although our method does not hold an absolute advantage in terms of speed, it still performs well overall. Our SeMv-3D method achieves an average inference time of approximately **46 seconds** per prompt, representing a **~29% speedup** compared to the current state-of-the-art prior-based model, Shap-E, which requires **65 seconds**. Furthermore, our method demonstrates superior semantic consistency and enables any-view generation.
>
>    In comparison to the previous **any-view** method, SPAD, which has an average inference time of **66 seconds**, our method achieves a **~30% speedup** while also exhibiting better multi-view consistency. These results highlight that SeMv-3D not only offers significant advantages in semantic and multi-view consistency but also achieves substantial improvements in efficiency.
>
> 2. Thank you for pointing this out. We have added the prompts in figure 5. Regarding the effectiveness of CA, as shown in the figure 5, SVS with no CA&OA results in surface blurriness, lack of detailed textures, and significant edge artifacts, such as at the foot of the bed on the right side of the figure. As illustrated in the middle column of the figure, **introducing CA mitigates these edge artifacts due to some semantic constraints**. It demonstrates that CA has some effect, but its impact is limited when used alone. Our proposed semantic alignment aims to **align the 3D visual features with semantic features** through CA and OA.

---

> > ### Comment · Reviewer_b6y7 · 2024-11-28
> >
> > ET3D is not an optimization-based method, which trains a generator distilled from MVDream and inference any objects within 0.008s. It uses the similar 3D representation, triplane, with your work.

---

> > > ### Author Response · Authors · 2024-11-28
> > >
> > > Thank you for pointing this out and clarifying the details of the method. We greatly appreciate your insightful feedback, which has significantly improved our understanding.
> > >
> > > Initially, we categorized models trained with distillation as optimization-based approaches. However, after revisiting the method and carefully considering your comments, we acknowledge that it can indeed be classified as a general text-to-3D method, aligning with the scope of our work. We apologize for any oversight in our initial interpretation.
> > >
> > > Regrettably, due to the lack of an open-source implementation, we are unable to perform a direct comparison. Based on the results presented in the paper, the method does appear to outperform our approach in certain aspects.
> > >
> > > While we recognize the impressive results demonstrated in the paper, we also have some thoughts regarding potential issues with the ET3D. Specifically, it employs MVDream as the teacher model, which we have shown to exhibit inconsistencies across different views. A natural question arises as to whether these inconsistencies are inherited by the student model trained through distillation. However, as the method is not currently open-sourced, we are unable to empirically test this hypothesis or conduct a comprehensive evaluation.
> > >
> > > Nevertheless, we fully agree that comparing with this method would be a valuable addition to our work. If the implementation becomes available in the future, we will prioritize including it in our experiments to provide a more thorough evaluation.
> > >
> > > Once again, we sincerely thank you for your valuable suggestion, which has significantly contributed to refining the scope and rigor of our work.

---

> > > > ### Comment · Reviewer_b6y7 · 2024-11-29
> > > > **Disagree with the multi-view reconstruction-based methods should be excluded in related works**
> > > >
> > > > From the inference and application aspect, the mentioned instant3D/LGM/GRM can direct output 3D representation only text input. In particular, Instant3D and this work has similar training overhead, both need to finetune a T2I diffusion model and a DINO to extract visual feature. The major difference is fine-tuned diffusion model producing multi-view images in instant3D while triplane prior in this work. Even they both use triplane as final 3D representation. However, Instant3D outperforms this work on texture quality and open-world generalization.
> > > >
> > > > I am very doubt the effectiveness of proposed triplane prior in solving fast text-to-3D generation task.
> > > >
> > > > Additionally, this work has much more training overhead than fintuning-based methods (MVDream, DreamView), which only finetune a T21 diffusion model. n other words, they are still just text-to-image generative models. That' why I think the main experiments is comparison with  fintuning-based methods is unfair. Actually, text-to-3D generation is mainly divided into optimization-based, prior-based methods and multi-view reconstruction-based. I disagree the  fine-tuning-based methods is a common category in text-to-3D area. In DreamView, their 3D results are based on SDS optimization.

---

> > > > > ### Comment · Reviewer_b6y7 · 2024-11-29
> > > > > **There are no obvious effect of CA and OA for semantic consistency in Figure 5**
> > > > >
> > > > > As mentioned in Weakness 1.2, I still don't see the proposed CA and OA's obvious effectiveness for semantic consistency in Figure 5. The proposed CA and OA can slightly improve the texture quality and decrease blur. In my observation,  textual-visual
> > > > > alignment is mainly from the DINO feature, which prior-based methods lack, as claimed in line 256. However, using the DINO feature cannot be regarded as a contribution to this work. At least, authors refuse to compare with other text-to-3D methods which also use DINO to enhance semantic feature.
> > > > >
> > > > > In summary, I think the experiments cannot satisfy the claimed motivation in this work. And authors didn't answer my question directly and even gave a response that didn't match the facts (ET3D, latency of Shap-E). I tend to decrease the score.

---

> > ### Comment · Reviewer_b6y7 · 2024-11-29
> > **Shap-E reports 12s per prompts on V100 without image condition in their paper**
> >
> > In my experiments, it takes around 20s to output 20 rendering images and mesh per prompt on A100. Please report the inference setting for a fair comparison. I don't know why there is a huge gap between the reported 65s and 10-20s. Maybe the CPU has been allocated for other tasks.

---

### Meta-Review · Area_Chair_yqdi · 2024-12-19

**Metareview:**

This paper received mixed reviews, three borderline reject and two borderline accept. The reviewers raised issues about the novelty and motivations, the effectiveness of the triplane prior and training overhead. Moreover, reviewer vfgq the evaluation benchmark is not standard  and the experiments should be conducted on T3Bench. During the rebuttal, the authors have effectively addressed two reviewers' concerns but did not fully address the comments of reviewers vfgq and b6y7. Since these questions cannot be fully addressed and the authors did not provide additional feedback, AC adopted the recommendation of the reviewer b6y7.

**Additional Comments On Reviewer Discussion:**

During the rebuttal period, the authors have clarified technical details and explained why some baselines were not adopted.
However, reviewer b6y7 mainly focuses on the novelty and contributions of the proposed components as well as training overhead while reviewer vfqg raised the evaluation is not standard. These concerns have not been answered by the authors. Therefore, AC conclude the final rating.

---

### Decision · Program_Chairs · 2025-01-22

Reject